# Interrelationship between Share of Women in Parliament and Gender and Development: A Critical Analysis

**Subrat Sarangi [1,*], R. K. Renin Singh [2] and Barun Kumar Thakur [3]**

1    Business Management, MICA, Ahmedabad 380058, Gujarat, India
2    Chitkara Business School, Chitkara University, Rajpura 140401, Punjab, India
3    Department of Economics, FLAME University, Pune 412115, Maharashtra, India
*    Correspondence: r12023@astra.xlri.ac.in

**Abstract:** Gender and development are among the two most important components of any economy to sustain its perpetual and sustainable economic growth in both the long as well as short run. The role of women in parliament and the interrelationship between gender and development is critically analysed. Women's representation in parliament is the dependent variable and the predictor variables considered are gender development index, female access to assets, female labour force, and country GDP per capita. Data were collected from the UNDP human development report for the period 2015 to 2021–2022 and World Bank for 188 countries of which finally 159 were considered to develop the model based on data availability. We have used the theoretical lens of social stratification theory and gender role theory to frame the hypothesis. A random effects model-based panel regression analysis of the data indicated a strong positive relationship between gender development index and the dependent variable, but no relationship between female labour force, and access to assets. The study addresses a critical gap in policy and development of the literature on gender, politics, and development using a global data set, establishing the importance of indicators such as gender development index, and laying down the path for future research on the subject.

**Keywords:** gender development index; access to assets; politics; women's representation in parliament; female labour force participation

## 1. Introduction

The fifth sustainable development goals (SDGs) advocated to achieve gender equality and ensures "to achieve gender equality and empower all women and girls" and end the discrimination, inequality, violence, and any harmful practices against women and girls by 2030 (UNDP 2022). Despite the given importance of achieving gender equality in millennium development goals (MDGs) and SDGs, the reality has been a different story altogether (Sandhu 2022). Historically, "Politics being the domain of men" was an established belief for centuries; women needed to contest against them (Chafetz and Dworkin 1987; Roper and Tosh 2021). In the recent times, media visibility has played a role in the dominance of masculine traits associated with politics favouring men over women, which is a severe democratic problem (Thesen and Yildirim 2023). Earlier literature reflects a lower representation of women in politics (Nowotny et al. 1981) to the last few decades with a marked increase in women representation in political offices (Hessami and da Fonseca 2020) development and policy level changes have been evident. However there is still a dearth of gender equality at the economic and social levels at large (Stockemer and Byrne 2012; Zhu and Chang 2019; Lari et al. 2022). The lower representation of women in parliament has been mainly attributed to family and motherhood (Grechyna 2022); however, societal and media coverage (Thesen and Yildirim 2023) on the under-representation of women in politics (Dal Bó et al. 2017; Paxton et al. 2007) and women's political participation and voice (Ladam et al. 2018) has gained global momentum over the past decade. It has also drawn the attention of academia and has been the motivation for this research study.

The representation of women in parliament could be impacted due to culture, politics, and development, as well as the contribution of women to a nation's economy. Gender is used for women, men, and boys (Lorber 2001). Gender inequality is to determine whether there is gender equality or inequality in rights, responsibilities, economic participation and opportunities, educational attainment, health and survival probabilities, and political empowerment (Jayachandran 2015). Women leaders bring a difference to decision making, group attitude, and initiation of priority to underprivileged populations (Bertrand et al. 2019). The symbolic representation of women in politics emphasizes female role models and how they shape people's attitudes and behaviours (Barnes and Burchard 2013). Recent research on feminine traits in political parties not only resulted in lowering the level of partisan hostility but improved the behavioural approach of the political party toward out-party women politicians and parliamentarians (Adams et al. 2022). Thin and thick representation is used for women's representation in politics. Thin representation is the presence of women in parliament and assemblies (Tremblay 2007), and thick representation is the presence of women parliamentarians in any ministerial office (Wahman et al. 2021).

Women senators present better results than men in parliament (Burns et al. 2021). Female leaders are the new influencers for adolescents and young girls (Campbell and Wolbrecht 2006). Ethiopia has been a frontrunner among the African continent in improving the well-being of women and adolescent girls through policy and implementation programs (Jones et al. 2018). A study based in Qatar (Lari et al. 2022) established the preference for men in leadership positions than women, and more so for respondents never having attended school. These dynamics positively influenced women to enter politics (Ladam et al. 2018), and women became more vocal when they saw more women contesting the election and winning the same (Burns et al. 2021). There should be a policy to encourage new mothers to participate in political programs and participation (Shore 2020), as education increases women's participation in politics (Grechyna 2022), and encouraging mothers indirectly increases the political participation of young girls (Arvate et al. 2021). Female representing higher positions positively affects women in the professional arena too to build their opinion toward politics (Ladam et al. 2018), irrespective of the win or loss in the elections (Sulkin 2005). These have been some of the research evidence established on the positives of increased involvement of women in politics.

There have been difficulties with expectations for differences in attitudes and behaviours between males and females, particularly when it reveals the commonalities between the genders (Cammisa and Reingold 2004). It depends a lot on the choices that are available to them when it is seen as the result of a failure to view politics from a gendered perspective and the impact of gender on the social context (Schneider and Bos 2019; Crawford 1995; Lovenduski 1998a). Some of the research on gender inequalities in politics is due to: (a) female willingness to run for a candidate position (Fox and Lawless 2004; Júlio and Tavares 2017; Schlozman et al. 1994); (b) political party selection of candidates: parties tend to select men more than women because the likelihood of men winning is higher (Kunovich and Paxton 2005); (c) voter's selection of candidates: the voters might be gender biased where they tend to cast a vote to a male candidate (Schwindt-Bayer 2010); and (d) electoral rules: the proportional system is better for promoting men in electoral politics (Iversen et al. 2010).

Development and related goals have always remained an area of critical research for academicians. At a holistic level, it has been observed that regional development within a country is driven by ethnic entrepreneurs and not as much by migrant entrepreneurs (Sarangi et al. 2022). From the perspective of women's representation in parliament and its associated effects on gender and developmental issues, the research is not just scarce but to gain momentum at the global scale. Given the above background, the primary research objective of this study is to understand the interrelationship between women representation in parliament with gender and development across countries studied globally. For this, we have collected data on gender development index (GDI) along with other variables, such as, female labour force participation (FLF), access to assets, and gross domestic product per

capita (GDP) from the UNDP reports (between 2015 and 2022) and World Bank. Female labour force and gender development index are important factors in understanding overall development issues (Nagata 2017; Choudhury et al. 2020; Rani and Kumar 2021). We have used the social stratification theory and gender role theory to frame the hypothesis and further tested the same using multiple regressions.

While over the past one and half decades there is a significant amount of research in the area at specific country or a regional level, we focus on a global data set and analyse the relationship between women representatives in parliaments and its associated effects on gender and development. This paper is structured into six sections. The first is the introduction section, which is followed by the literature review, critically examining the body of knowledge on gender and development issues. The third section emphasizes the theoretical framework followed by the sub sections on data and variables, hypothesis development, and methodology. It is then followed by the results and we discuss the findings in the Discussion section. The paper ends with the Conclusion section followed by references.

## 2. Literature Review

Historically, stereotype masculine traits have been the preferred trend reflected in political literature over feminine traits (Alexander and Andersen 1993; Rosenwasser and Dean 1989; Sanbonmatsu 2002), which is gradually changing and have started showing support for feminine traits and public acceptance (Cormack and Karl 2022). Power is the central problem of politics, politics results from abuses of power, and politics is nothing else but the power struggle (Minogue 1959; Buchanan and Badham 2020), concerns controlling people and events, which also extends to the psychological state of influencing others (Anderson et al. 2012; Bugental et al. 1989). Politicians' influence is the key to influencing the masses or even getting any policies approved in the legislative assemblies or parliaments (Besley and Reynal-Querol 2011). It is not only restricted to framing policies but also implementation (Dal Bó et al. 2017) and includes the thought process of other individuals, their emotions, and the actions conducted by an individual or group of people, extending their well-being (Boehm et al. 1993; Simpson and Willer 2015).

The combination of power with masculine traits in politics and political parties made politics a gendered organization (Lovenduski 2005). Individual belief can be based on individual sources of power, position power, or interpersonal relationships (Bugental et al. 1989; Bugental and Lewis 1999). The rapid increase in women's access to power in politics and women's appointment at high-prestige posts have changed the rules of the power game in politics (Arriola and Johnson 2014; Escobar-Lemmon and Taylor-Robinson 2009) and even mitigate cross-party hostility (Adams et al. 2022). The recent example has been the head of the state in India—Honourable President of India, Ms. Draupadi Murmu serving from 25 July 2022, representing the female gender and belonging to the tribal community from the eastern state of Odisha. Thus, the inclusion of women in a power positions in politics such as defence, foreign affairs, and finance has led to women's ascension to other such portfolios in politics and parliaments, altering traditional gendered governance patterns (Barnes and O'Brien 2018).

During the post-World War II period, women were entitled to political rights without much resistance (Paxton et al. 2006; Rupp and Taylor 1999). The scenario of women in politics was in the nascent stage then. By the end of the 20th century, there was a considerable increase in the candidature of women for political participation in western countries (Wasburn and Wasburn 2011). Despite the increase in women's presence in the political system, it was still dominated by masculine traits until the early 2000s (Dolan 1997; Sanbonmatsu 2002). Since then, there has been a sea change in the representation of women in electoral politics (Cormack and Karl 2022). The presence of women in the ecosystem of political and ministerial office makes other women feel more connected to the political system (Reingold and Harrell 2010), and this rise in female empowerment usually leads to the proper allocation of resources and reduced corruption (Jha and Sarangi

2018). Female capabilities result in the economic and political change (Hossain et al. 2019). Women's political empowerment considers civil liberties, civil society participation, and political participation for women in the index (Alexander et al. 2016; Sundström et al. 2017). Access to health and education is directly associated with women's political empowerment, broadening human capital (Hornset and de Soysa 2022). Developed nations have seen a considerable increase in the participation of women in politics and political activities compared to developing and underdeveloped nations. The above discussions highlight the multiple antecedents causing the changes in willingness, experience, and electoral systems as documented in the extant literature (Arceneaux 2001; Durante et al. 2013; Paxton and Kunovich 2003).

Globally there is very little representation of women in legislative bodies compared to men. The main reason behind this is the more time devoted to the family (Chhibber 2002). Gender inequality concerns are mainly due to literacy rate and level of education (Arceneaux 2001; Johnson et al. 2003). However, it is to be noted that the presence of female politicians and parliamentarians enhances the confidence, trust, and satisfaction of female citizens (Karp and Banducci 2008) and engendering political engagement among females (Barnes and Burchard 2013). However, literature on a meta-analysis on gender differentiation strongly supported male politicians (Van der Pas and Aaldering 2020). Failure of political parties across the countries to meet gender quotas and the greater presence of females (Dahlerup and Freidenvall 2005), formal and informal rules (Chappell 2006), decision making due to political elites deciding on behalf of the masses, which differs systematically from the masses (Kertzer 2022) are some of the other dominant areas of research on the theme. Specific to the political recruitment and selection process of candidate's interest in gender politics (Kenny 2013), symbolic effects of women's collective representation (Adams et al. 2022), gender and affective polarization (Klar 2018; Ondercin and Lizotte 2021), party gender composition (Dolan 1997; Rashkova 2021), and gender political socialization (Bos et al. 2022) have been some of the recent phenomena that have engaged academia and need to explore it further.

An analysis of women representation in high political positions in recent years indicates an increase in Europe, East Asia, Sub-Saharan Africa, and The Pacific countries. There has been only one female head of state in North America in the past 50 years. The longest-standing female leaders have ruled over Iceland for 16 years and 16.1 years over Germany and Dominica for 14.9 years, compared to 14 years in Ireland[1]. The gender gap is prominent in the political arena. According to World Economic Forum in 2017, there is around a 23% gender gap in politics globally. This makes the theme interesting and motivates us to delve deeper into the subject at a global level.

Males were less inclined than females to support women standing for the Senate, irrespective of the political party (Cook 2019). White men have stopped supporting feminism and the women's movement since the last quarter of the 20th century (Hansen 1997). Economic considerations may compel some women to choose positions that need less training, but qualifications for a prominent position may necessitate the development of a career (Ragins and Sundstrom 1989) and women entering the labour market (Choudhury et al. 2020). In Latin-American cabinets, presidents opted for men and women in a different types of ministries—women are given cabinet posts of lower prestige (Escobar-Lemmon and Taylor-Robinson 2005; Morgan and Buice 2013). There is the belief that leadership is a masculine trait; women who are leaders often encounter bias, and women run into difficulty because people view their dictatorial behaviour differently than they perceive that of men (Eagly et al. 1992). There is a conscious question if women make a difference in policies by prioritizing different political issues, voting differently, making changes in bills, and the effectiveness with which they would try to pass a bill (Paxton et al. 2007), why is there a low acknowledgment by people? The individual in a leadership position is high on self-confidence, achievement, and dominance, and society relates these characteristics with a male role more (Kelly 1983; Rosenwasser and Dean 1989). In an Italian study on fifteen men and an equal number of women, it was found that men are mostly involved

in employment in the public sphere, whereas women are more in family chores, and this gender hierarchy defines the boundaries of symbolic behaviours and objects associated with masculinity and femininity (De Simone et al. 2018). Females trying to establish themselves in the market are forced to position themselves as contenders against their male counterparts and must play the game with the mindset of not perishing out in the rules set by society (Kanze et al. 2018). If the success of the company is unaffected by having more female directors than males who are equally qualified, they should be given preference when decisions about promotions are made (Pletzer et al. 2015).

The impact of the increased presence of women in active politics at a global level shall have a profound influence on several indicators (Funk et al. 2021; Lv and Deng 2019). It would result in an increase in the country's per capita GDP on the global average (Stockemer and Byrne 2012). It will draw a greater participation of adolescent girls in politics as the more prominent female role models come into world politics (Arvate et al. 2021; Campbell and Wolbrecht 2006). In developing and poor nations, women's engagement in the labour participation force may have distinct effects or interactions with cultural aspects of the role of women in political representation (Stockemer and Byrne 2012). Women's participation in the workforce enhances their representation in the parliament, and an increase in women's labour force participation has a significant beneficial impact on the future growth of the nation (Ragins and Sundstrom 1989). Therefore, to foster development and growth, gender diversity should be encouraged with fairness. Therefore, in this paper, we examine the role of women in politics and its influence on gender and development.

## 3. Theoretical Framework, Data and Methodology

To investigate the role different socio-economic factors play in female participation in parliament, the study uses social stratification theory and gender role theory. Social stratification theory posits that factors such as inequalities due to economic class, social class, race and gender influence an individual's opportunities and outcomes. Individuals who are situated at higher positions in the hierarchy have access to power and resources than those who are situated at lower positions (Grabb 1990). Gender role theory suggests that gender roles are not based on biological sex but are instead influenced by social and cultural factors. The theory posits that any individual can have both masculine and feminine traits, but they learn these gender roles through socialization. Further, these roles can influence their behaviour and attitude in various domains (Bem 1974; Yarnell et al. 2019; Roper and Tosh 2021).

Studies have used the gender role theory (Diekman and Schneider 2010; Schneider and Bos 2019) to suggest that gender stereotypes and societal expectations of women's role lead to lower representation and success of women in politics (Carroll and Sanbonmatsu 2009; Fox et al. 2001). Building on the theories viz. social stratification theory and gender role theory, we develop the hypothesis discussed in Section 3.2.

### 3.1. Data and Variables

The study uses share of women in parliament (WRP) as the dependent variable. The main independent variables in the equation are gender development index (GDI), female labour force (FLF) participation, and access to assets (ATA). To account for potential confounding effect of standard of living of various countries, the gross domestic product per capita has been included in the equation as a control variable. The data for gender development index was collected from human development report (HDR) from the years 2015 to 2022 published by United Nations Development program (UNDP). The data for share of women in parliament, GDP per capita, access to asset and labour force participation was collected from World Bank database. Data were collected for 188 countries as shown in Table 1.

**Table 1.** List of countries. https://eige.europa.eu/gender-equality-index/2022/EU (accessed on 10 January 2023).

| Sl. No. | Country | Sl. No. | Country | Sl. No. | Country | Sl. No. | Country |
|---|---|---|---|---|---|---|---|
| 1 | Norway | 51 | Russian Federation | 100 | Tonga | 150 | Eswatini |
| 2 | Australia | 52 | Oman | 101 | Belize | 151 | Taiwan |
| 3 | Switzerland | 53 | Romania | 102 | Dominican Republic | 152 | Nigeria |
| 4 | Denmark | 54 | Uruguay | 103 | Suriname | 153 | Cameroon |
| 5 | Netherlands | 55 | Bahamas | 104 | Maldives | 154 | Madagascar |
| 6 | Germany | 56 | Kazakhstan | 105 | Samoa | 155 | Zimbabwe |
| 7 | Ireland | 57 | Barbados | 106 | Botswana | 156 | Mauritania |
| 8 | United States | 58 | Antigua and Barbuda | 107 | Moldova | 157 | Solomon Islands |
| 9 | Canada | 59 | Bulgaria | 108 | Egypt, Arab Rep. | 158 | Papua New Guinea |
| 10 | New Zealand | 60 | Palau | 109 | Turkmenistan | 159 | Comoros |
| 11 | Singapore | 61 | Panama | 110 | Gabon | 160 | Yemen |
| 12 | Hong Kong SAR, China | 62 | Malaysia | 111 | Indonesia | 161 | Lesotho |
| 13 | Liechtenstein | 63 | Mauritius | 112 | Paraguay | 162 | Togo |
| 14 | Sweden | 64 | Seychelles | 113 | Palestine, State of | 163 | Haiti |
| 15 | United Kingdom | 65 | Trinidad and Tobago | 114 | Uzbekistan | 164 | Rwanda |
| 16 | Iceland | 66 | Serbia | 115 | Philippines | 165 | Uganda |
| 17 | Korea, Rep. | 67 | Cuba | 116 | El Salvador | 166 | Benin |
| 18 | Israel | 68 | Lebanon | 117 | South Africa | 167 | Sudan |
| 19 | Luxembourg | 69 | Costa Rica | 118 | Vietnam | 168 | Djibouti |
| 20 | Japan | 70 | Iran, Islamic Rep. | 119 | Bolivia | 169 | South Sudan |
| 21 | Belgium | 71 | Venezuela, RB | 120 | Kyrgyz Republic | 170 | Senegal |
| 22 | France | 72 | Turkiye | 121 | Iraq | 171 | Afghanistan |
| 23 | Austria | 73 | Sri Lanka | 122 | Cabo Verde | 172 | Cote d'Ivoire |
| 24 | Finland | 74 | Mexico | 123 | Micronesia (Federated States of) | 173 | Malawi |
| 25 | Slovenia | 75 | Brazil | 124 | Guyana | 174 | Ethiopia |
| 26 | Spain | 76 | Georgia | 125 | Nicaragua | 175 | Gambia |
| 27 | Italy | 77 | St. Kitts and Nevis | 126 | Morocco | 176 | Congo, Dem. Rep. |
| 28 | Czechia | 78 | Azerbaijan | 127 | Namibia | 177 | Liberia |
| 29 | Greece | 79 | Grenada | 128 | Guatemala | 178 | Guinea-Bissau |
| 30 | Estonia | 80 | Jordan | 129 | Tajikistan | 179 | Mali |
| 31 | Brunei Darussalam | 81 | North Macedonia | 130 | India | 180 | Mozambique |
| 32 | Cyprus | 82 | Ukraine | 131 | Honduras | 181 | Sierra Leone |
| 33 | Qatar | 83 | Algeria | 132 | Bhutan | 182 | Guinea |
| 34 | Andorra | 84 | Peru | 133 | Timor-Leste | 183 | Burkina Faso |
| 35 | Slovakia | 85 | Albania | 134 | Syrian Arab Republic | 184 | Burundi |
| 36 | Poland | 86 | Armenia | 135 | Vanuatu | 185 | Chad |
| 37 | Lithuania | 87 | Bosnia and Herzegovina | 136 | Congo, Rep. | 186 | Eritrea |
| 38 | Malta | 88 | Ecuador | 137 | Kiribati | 187 | Central African Republic |
| 39 | Saudi Arabia | 89 | Saint Lucia | 138 | Equatorial Guinea | 188 | Niger |

**Table 1.** *Cont.*

| Sl. No. | Country | Sl. No. | Country | Sl. No. | Country | Sl. No. | Country |
|---|---|---|---|---|---|---|---|
| 40 | Argentina | 90 | China | 139 | Zambia | | |
| 41 | United Arab Emirates | 91 | Fiji | 140 | Ghana | | |
| 42 | Chile | 92 | Mongolia | 141 | Lao People's Democratic Republic | | |
| 43 | Portugal | 93 | Thailand | 142 | Bangladesh | | |
| 44 | Hungary | 94 | Dominica | 143 | Cambodia | | |
| 45 | Bahrain | 95 | Libya | 144 | Sao Tome and Principe | | |
| 46 | Latvia | 96 | Tunisia | 145 | Kenya | | |
| 47 | Croatia | 97 | Colombia | 146 | Nepal | | |
| 48 | Kuwait | 98 | Saint Vincent and the Grenadines | 147 | Pakistan | | |
| 49 | Montenegro | 99 | Jamaica | 148 | Myanmar | | |
| 50 | Belarus | | | | | | |

Gender Development Index (GDI): In 1995, the United Nations Development Program (UNDP) introduced the gender development index (GDI) to incorporate a gender-sensitive perspective into the human development index (HDI). The GDI measures gender inequalities in three areas: life expectancy, education, and income.

Female Labour Force participation (FLF): It is defined as the percentage of women (15 years or more) who are actively engaged in paid employment or are seeking employment opportunities in the labour force.

Access to Assets (ATA): It measures if the law entitles men and women with equal ownership of immovable property.

Gross Domestic Product (GDP) per capita: Gross domestic product per capita of a country is an indicator of standard of living of people in a country. It is calculated my measuring the total economic output of a country divided by the total number of people living in the country. It has been used as a control variable in the model.

Percentage of Women Representation in Parliament (WRP): The variable determines the share of seats in parliament in a country held by women. It serves as the dependent variable in the study.

*3.2. Hypothesis Development*

Gender Development Index and Women Representation in Parliament: GDI comprises life expectancy, education, and income. Based on the social stratification theory, it can be hypothesized that women who have higher levels of education and income have a higher chance of actively participating in politics as they will have higher access to political resources such as information and network. Further, women who have limited access to these political resources are less likely to take part in politics. Literacy and education are central to social upliftment and economic development (Moghadam and Senftova 2005) and are central to attaining the millennial development goals. Studies have found influence of education (Gleason 2001) and socio-economic status (Lovenduski 1998b; Ballew et al. 2020) on women empowerment and female political participation. Thus, we bring forth the following hypothesis:

**H1.** *There is a significant positive relationship between GDI on percentage of women representation in parliament.*

Access to Assets and Women Representation in Parliament: A country where the law provides unequal access to assets based on gender is creating a social stratum. The social stratification theory suggests that women in countries where the law provides access to assets will allow women to channel these resources to pursue their political ambitions. On the contrary, in countries where the law limits the access to resources will have lower participation in politics by women. Khayyam and Tahir (2019) in their study found that

the major obstacle towards female participation in the political arena has been the social structure developed by a male dominated society. Vissandjée et al. (2006) found that women limited social mobility is a key factor affecting persistently low level of female political participation. Studies suggest that the level of resources provisioned by the government (Gleason 2001). Thus, we formulate the following hypothesis:

**H2.** *There is a significant positive relationship between access to assets on percentage of women representation in parliament.*

Female Labour Force participation and Women Representation in Parliament: Based on the gender role theory, it can be hypothesized that countries where women have higher labour force participation there will be a greater role for women in jobs, and hence have a higher propensity to participate in politics. On the other hand, women who must adhere to traditional gender roles are less likely to participate in politics. Social roles have established that men specialize in agriculture work outside the home while women specialize in activities within the home, and this belief persists even outside of an agricultural economy and include entrepreneurship and politics (Giuliano 2014). Gleason (2001) found female labour force participation, household obligations and history of acceptance of women in roles of political power to be important determinants of female candidacy for the state legislature. Thus, the following hypothesis is suggested:

**H3.** *There is a significant positive relationship between female labour force participation on percentage of women representation in parliament.*

### 4. Methodology

EViews 9.0 software was used to conduct the analysis. A panel data regression model using one-way random-effects model was used. Women representation in parliament is regressed with the set of independent variables i.e., gender development index (GDI), female labour force (FLF) participation, access to assets (ATA), and GDP per capita (used as control variable) as shown in Equation (1).

$$WRP_{it} = \beta_0 + \beta_1 GDI_{it} + \beta_2 FLF_{it} + \beta_3 ATA_{it} + \beta_4 GDP_{it} + \omega_i + \varepsilon_{it} \tag{1}$$

where

$WRP_{it}$ is the dependent variable for the *i*-th entity (cross-sectional unit) at time *t*,

$GDI_{it}$, $FLF_{it}$, $ATA_{it}$, and $GDP_{it}$ are the independent variables for the *i*th country at time *t*,

$\beta_0$ is the intercept,

$\beta_1$, $\beta_2$, $\beta_3$, $\beta_4$ are the coefficients for the independent variables,

$\omega_i$ is the random effect that captures unobserved heterogeneity across countries that is constant over time,

$\varepsilon_{it}$ is the idiosyncratic error term. The summary statistics of the variables for all the countries together are presented in Table 2. The data was collected for a total of 188 countries for the years 2014, 2015, 2017, 2018, 2019, and 2021. It may be noted that data for 2016 and 2020 was not included as the same was not published by the UNDP (our source for the global data set). The number of countries included in the final model was 159 due to missing data (from a total of 188 countries). Further, the model was unbalanced as data for some years for certain countries were missing.

**Table 2.** Summary Statistics.

|  | **WRP** | **GDP** | **GDI** | **FLF** | **ATA** |
|---|---|---|---|---|---|
| Mean | 23.01 | 15,497.55 | 0.94 | 50.71 | 83.03 |
| Maximum | 63.75 | 133,590.10 | 1.00 | 83.47 | 100.00 |
| Minimum | 0.00 | 216.97 | 0.30 | 10.92 | 0.00 |
| Std. Dev. | 11.99 | 21,108.08 | 0.07 | 13.60 | 25.32 |
| Observations | 892.00 | 892.00 | 892.00 | 892.00 | 892.00 |
| VIF |  | 1.15 | 1.56 | 1.25 | 1.64 |

The data were tested for multicollinearity. The variance inflation factor (VIF) values for the four variables were less than 10 indicating no multicollinearity (See Table 2). The correlation values among the independent variables are presented in Table 3. Further, the data was tested for heteroskedasticity, and the Breusch–Pagan value was found to be 5.76, with 4 degrees of freedom and *p*-value of 0.21. Therefore, we do not reject the null hypothesis of absence of heteroscedasticity present in the residuals of the linear regression model.

**Table 3.** Correlation between variables.

|  | **GDI** | **FLF** | **ATA** | **GDP** | **C** |
|---|---|---|---|---|---|
| GDI | 29.78 | −0.01 | −0.04 | 0.00 | −24.12 |
| FLF | −0.01 | 0.00 | 0.00 | 0.00 | −0.10 |
| ATA | −0.04 | 0.00 | 0.00 | 0.00 | −0.02 |
| GDP | 0.00 | 0.00 | 0.00 | 0.00 | 0.00 |
| C | −24.12 | −0.10 | −0.02 | 0.00 | 29.87 |

Further, Hausman test was conducted to check the appropriate model for the analysis. The test suggested that random effect model will be more suitable as compared to fixed effect model. The output of random effect panel regression model is presented in Table 4.

**Table 4.** Regression results.

| **Variable** | **Coefficient** | **Std. Error** | **t-Statistic** | **Prob.** |
|---|---|---|---|---|
| C | −9.74 | 7.99 | −1.21 | 0.2235 |
| GDI | 17.13 | 5.78 | 2.95 | 0.0032 * |
| FLF | 0.17 | 0.09 | 1.86 | 0.0626 |
| ATA | 0.06 | 0.05 | 1.16 | 0.2433 |
| GDP | 0.00016 | $4.73 \times 10^{-5}$ | 3.45 | 0.0006 * |

Note: * Statistically significant at 0.05.

The model has an F-statistic of 57.5 and is statistically significant. The adjusted R square value is 0.911 indicating that the independent variables in the regression model are good predictors of the dependent variable (i.e., women representation in parliament).

All the predictor variables (i.e., GDI, FLF, and ATA have a positive relationship with the dependent variable (WRP), while only GDI is the only significant predictor with a partial correlation coefficient of 17.13. As GDP was used as a control variable in the equation, it suggests that even though it is a significant variable, it may not be an important factor in explaining the variation in the dependent variable.

Based the results of the analysis, the final regression equation is presented below:

$$WRP_{it} = -9.74 + 17.13GDI_{it} + 0.17FLF_{it} + 0.06ATA_{it} + 0.00016GDP_{it} + \omega_i + \varepsilon_{it} \quad (2)$$

## 5. Discussion

The study investigated the relationship between the independent variables (gender development index, female labour force participation, and access to assets) on the dependent

variable (women representation in parliament). The findings of the study are discussed subsequently in detail (refer Table 5 for result summary).

**Table 5.** Summary of hypothesis testing.

| Hypothesis | Description | Variables | Results | Conclusion |
|:---:|:---|:---:|:---:|:---:|
| H1 | There is a significant positive relationship between GDI on percentage of women representation in parliament. | GDI and WRP | $\beta = 17.12$, $t(789) = 2.95$, $p < 0.05$ | Supported |
| H2 | There is a significant positive relationship between access to assets on percentage of women representation in parliament. | ATA and WRP | $\beta = 0.06$, $t(789) = 1.16$, $p > 0.05$ | Not supported |
| H3 | There is a significant positive relationship between female labour force participation on percentage of women representation in parliament. | FLF and WRP | $\beta = 0.17$, $t(789) = 1.86$, $p > 0.05$ | Not supported |

The first hypothesis proposed that GDI will have a significant and positive relationship with women representation in parliament. The results show that the hypothesis is supported ($\beta = 17.13$, $p < 0.05$) validating the findings of Hessami and da Fonseca (2020), Lovenduski (1998a), Ballew et al. (2020), and Hornset and de Soysa (2022). The result is consistent with social stratification theory, which suggests that countries with higher GDI will have higher representation of women in parliament and power. Our study establishes the linkage at a global level with data from 188 countries.

The second hypothesis proposed that FLF will have a significant and positive relationship with women representation in parliament. The result was counterintuitive ($\beta = 0.17$, $p > 0.05$), and inconsistent with gender role theory, which suggests that countries with higher participation of women in labour force will have a higher WRP. One possible reason for the inconsistency could be that factors such as political climate of a country (de Mesquita and Smith 2011), cultural fabric (Schneider and Bos 2019), women's standing in society (Chattopadhyay and Duflo 2004; Jha and Sarangi 2018), and attitude towards women in political roles (Ahmed and Moorthy 2021) might be more important determinants of the dependent variable. Further studies should examine the effect of these aspects on the representation of women in parliament.

The third hypothesis proposed that ATA will have a significant and positive relationship with WRP. However, the hypothesis was not supported ($\beta = 0.17$, $p > 0.05$) and the finding is inconsistent with social stratification theory, which suggests that countries providing women with access to assets will have a higher representation of women in parliament. This could have happened because of confounding variables such as attitude towards women's political leadership (Eto 2013), and political party structure (Kunovich and Paxton 2005), to name a few. It remains to be investigated in greater detail in future studies.

## 6. Conclusions

Summarizing the findings, we observe that one of the three hypothesis is supported (i.e., WRP and GDI), while two of them are rejected (WRP and ATA, WRP and FLF). This has several theoretical and practical implications for policy making. From a theoretical perspective, the major implication is that our findings open the theme for further investigation, specifically the counterintuitive results of WRP and ATA, and WRP and FLF. The role of cultural factors related to perception of women in politics and power positions in the developed, developing, and underdeveloped countries needs to be specifically explored. In addition, the perception of political parties towards women candidature in the electoral process, evolution, and structure of political parties and inclusivity policies, and overall standing of women in the society need to be studied under the social stratification and gender role theories. The need to elevate women in society (Htun and Weldon 2018) to



build the 'institutional capability' (Sen 1999) for development is key. The future studies on the subject need to specifically investigate the factors in detail and establish a linkage with women empowerment in politics toward institutional capability building and development.

From a policy perspective, the study establishes a positive relationship between GDI and WRP. The determinants of GDI are life expectancy, education, and income. The extant literature also explains the enhanced role of women in development (Hossain et al. 2019; Jha and Sarangi 2018) and the inter relatedness of political institutions with development (Acemoglu and Robinson 2012). Hence, at a policy level, measures need to be adopted to raise GDI through education, gender parity in employment opportunities, protection of the girl child, and improved healthcare infrastructure for women to drive development at a national and global levels. This shall automatically set up the institutional capability for sustainable development.

There are some limitations of this study which we would like to highlight. Lack of data at a global level restricted the scope to include more independent variables and it remains an area of improvement for future. Second, while we started with a data set of 188 countries for an eight-year period, we could finally perform the analysis for 159 countries and a period of six years (as data were missing for 2016 and 2020). While we did explore other possible sources, but access to global data required for this type of study is scarce and limited. However, we strongly believe that the findings of this study shall benefit academicians and policymakers who are focused on politics, development, and gender-related studies significantly in the future.

**Author Contributions:** Conceptualization, S.S.; methodology, S.S. and R.K.R.S.; software, R.K.R.S.; validation, S.S., R.K.R.S. and B.K.T.; formal analysis, R.K.R.S.; investigation, R.K.R.S.; resources, R.K.R.S. and B.K.T.; data curation, R.K.R.S. and B.K.T.; writing—original draft preparation, S.S., R.K.R.S. and B.K.T.; writing—review and editing, S.S.; visualization, S.S. and R.K.R.S.; supervision, S.S.; project administration, S.S. All authors have read and agreed to the published version of the manuscript.

**Funding:** This research received no external funding.

**Institutional Review Board Statement:** Not applicable.

**Informed Consent Statement:** Not applicable.

**Data Availability Statement:** Publicly available datasets were analyzed in this study. This data can be found here: [https://eige.europa.eu/gender-equality-index/2022/EU].

**Conflicts of Interest:** The authors declare no conflict of interest.

## Note

1    https://eige.europa.eu/gender-equality-index/2022/EU (accessed on 10 January 2023).

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
