# Peer review of "Interrelationship between Share of Women in Parliament and Gender and Development: A Critical Analysis"

_admsci, doi:10.3390/admsci13040106_

Round 1

Reviewer 1 Report

Please find attached file

Reviewer 2 Report

Dear Authors,

 Your paper entitled ”Role of share of women in parliament and its impact on gender and development” treats a relatively current and interesting topic. However, the work has a number of shortcomings.

 The objective of the paper is not mentioned in Abstract.

 Many statements are not current and should be treated as such. Enormous progress has been made in recent years in terms of gender equality and women's representation in all areas. You state that ”Literature reflects a lower representation of women in politics (Nowotny et al., 1981)”. The source is from 1981. We are in 2023. How valid is this statement today?

 At the end of Introduction section, you wrote: ”Given the above background, the primary research objective of this study is: Does the share of women in parliament have any impact on gender and developmental issues?”. This may be the research question .

In the Introduction you specify that "we constructed the gender inequality index". It already exists. It is calculated by the UNDP.

In the Introduction, it would be advisable to develop your research motivation.

 In the Literature Review section old sources predominate. It would be advisable not to insert figures in the analysis of the theoretical framework.

 All abbreviations must be explained/detailed.

 In the Literature Review it would be advisable to analyse studies on the topic oriented by country groups, as you have designed the empirical part.

 The methods will be presented technically.

 The choice of indicators used must be justified. For example, in my opinion, it is not normal to analyse the representation of women in Parliament in relation to secondary education. I have no argument that secondary education is a factor that reduces gender inequality in Parliament. However, I think a much better option would be tertiary education.

 The period analysed is not at all representative.

 Why did you select "Adolescent birth rate" as an independent variable? GGI is also calculated based on the "Adolescent Birth Rate". I do not know to what extent GGI is a suggestive variable as long as it is calculated on the basis of other variables analysed. It would be appropriate to present the correlation table.

 The methods are briefly presented and applied. I suggest you to apply the methods step by step without skipping any regression assumptions, test for normality, correlations, explain all variables and insert regression equations based on the results.

 Discussions should be extended based on results and theory.

 The title of the paper is ”Role of share of women in parliament and its impact on gender and development”. If you want to demonstrate the impact of the share of women in parliament on gender and development" the share of women in parliament can only be an independent variable. So, the title of the paper needs to be changed or the whole analysis.

 The Conclusions do not apply.

 In conclusion, I suggest you to bring the paper to a much improved form.

Round 2

Reviewer 2 Report

The title needs to be corrected.

,,Different countries” is too vague.

The justification of the variables choice should be strengthened. The most recent references should prevail. Where you cite, for example, a source from the 1960s, reinforce it with one from the current year or last year so that it can be seen that it is still valid.

The regression equation must also be adapted to the results.

I recommend that you separate the Discussions from the Conclusions.
